# Ticks and Chlamydia-Related Bacteria in Swiss Zoological Gardens Compared to in Contiguous and Distant Control Areas

**DOI:** 10.3390/microorganisms11102468

**Published:** 2023-09-30

**Authors:** Vincent Vanat, Sébastien Aeby, Gilbert Greub

**Affiliations:** 1Institute of Microbiology, University of Lausanne and University Hospital Center (CHUV), 1005 Lausanne, Switzerland; vincent.vanat@unil.ch (V.V.); sebastien.aeby@chuv.ch (S.A.); 2Service of Infectious Diseases, University Hospital Center (CHUV), 1005 Lausanne, Switzerland

**Keywords:** chlamydia, *Chlamydiae* phylum, *Chlamydiales* order, *Rhabdochlamydia*, *Parachlamydia*, intracellular bacteria, chlamydia-like bacteria, Ixodes ticks, epidemiology, real-time PCR

## Abstract

Ticks are vectors of numerous agents of medical importance and may be infected by various *Chlamydia*-related bacteria, such as members of *Parachlamydiaceae* and *Rhabdochlamydiaceae* families, which are sharing the same biphasic life cycle with the pathogenic *Chlamydia*. However, the veterinary importance of ticks and of their internalized pathogens remains poorly studied. Thus, we wondered (i) whether the prevalence of ticks was higher in zoological gardens than in control areas with similar altitude, vegetation, humidity and temperature, and (ii) whether the presence of *Chlamydia*-related bacteria in ticks may vary according to the environment in which the ticks are collected. A total of 212 *Ixodes ricinus* ticks were collected, and all were tested for the presence of DNA from any member of the *Chlamydiae* phylum using a pan-*Chlamydiae* quantitative PCR (qPCR). We observed a higher prevalence of ticks outside animal enclosures in both zoos, compared to in enclosures. Tick prevalence was also higher outside zoos, compared to in enclosures. With 30% (3/10) of infected ticks, the zoological gardens presented a prevalence of infected ticks that was higher than that in contiguous areas (13.15%, 10/76), and higher than the control distant areas (8.65%, 9/104). In conclusion, zoological gardens in Switzerland appear to contain fewer ticks than areas outside zoological gardens. However, ticks from zoos more often contain *Chlamydia*-like organisms than ticks from contiguous or distant control areas.

## 1. Introduction

Ticks are obligate hematophagous arthropods. They are usually present in forests, shrubs, and tall grass. *Ixodes ricinus* is the most common tick species in Switzerland [1]. The geographical repartition of *Ixodes ricinus* has been shown to be strongly linked to environmental conditions. Thus, areas with dense vegetation, high temperature and high humidity appear to be especially favorable for *Ixodes ricinus* [2,3]. 

In the last few years, tick habitats have extended, especially into urban and suburban areas [4] and to higher altitudes [5]. Thus, from 2008 to 2018, the region suitable for ticks in Switzerland extended from 16% to 25% [5]. 

In Europe, the species *Ixodes ricinus* is responsible for the transmission of a number of major human pathogens, such as tick-borne encephalitis virus (TBEV), *Borrelia* spp., and *Anaplasma phagocytophilum* [6,7,8]. Recent studies have demonstrated that *I. ricinus* ticks collected from Switzerland can also be infected with bacteria belonging to the *Chlamydiae* phylum. Ticks may thus serve as a vector or even a reservoir for these intracellular bacteria, which are commonly isolated from environments such as water, or soil [9,10,11], and from animals such as mammals, birds, protists, insects and arthropods [12,13,14]. Thus, bacteria of the phylum *Chlamydiae* appear to be much more ubiquitarian than previously thought. 

The bacteria of the *Chlamydiae* phylum are often referred as *Chlamydia*-related bacteria, since they share the same biphasic developmental life cycle with infectious elementary bodies and dividing metabolically active reticulate bodies [15]. The *Chlamydiae* phylum is currently composed of nine family-level lineages, including *Chlamydiaceae*, *Clavichlamydiaceae*, *Criblamydiaceae*, *Parachlamydiaceae*, *Piscichlamydiaceae*, *Rhabdochlamydiaceae*, *Simkaniaceae* and *Waddliaceae* [16,17,18]. Members of *Chlamydiaceae*, *Clavichlamydiaceae*, *Criblamydiaceae*, *Parachlamydiaceae*, *Rhabdochlamydiaceae*, *Simkaniaceae* and *Waddliaceae* have all been associated with ticks, but the most common species detected in ticks belong to the *Rhabdochlamydiaceae* and *Parachlamydiaceae* families [9,19,20,21,22,23]. It is as yet unclear whether the *Chlamydia*-related bacteria isolated from *I. ricinus* ticks could cause human or animal diseases. However, it appears that ticks can at least transmit *Chlamydia*-related bacteria to humans, with the bacteria having been documented in skin biopsies at the site of tick bites [20].

The presence of enclosures with different biotopes and many different animal species in zoological gardens may lead to the reproduction of ticks, and potentially to the spread of tick-borne pathogens from enclosures by rodents present in the neighborhood [24,25,26,27,28]. Various species of tick have been collected in zoological gardens, including *Dermacentor variabilis*, *Rhipicephalus sanguineus*, *Amblyomma* spp., and *I. ricinus.* The latter is the tick species most frequently collected within European zoological gardens [24]. Moreover, several serological studies have revealed the presence of *Borrelia* spp., TBEV [29,30,31], and *Babesia* spp. [32,33] in captive animals in zoos. Bats [34], pigs [35], cats and dogs [36] have all been found to have been infected with *Chlamydia*-related bacteria, even when kept in captivity. However, there has been—until the present work—no evidence of *Chlamydia*-related bacteria in ticks from zoological gardens, mainly due to the total absence of such investigations. 

Interestingly, in the Zoo of Sao Paulo, the number of ticks collected on captive animals was shown to be lower than the number of ticks collected on free-living animals in the zoological park [37]. However, such studies in zoos are rare, and very few studies have compared the number of ticks present in enclosures to those in control areas outside the enclosures or even outside the zoos, and none have looked specifically at *Chlamydiae* from ticks. 

Thus, we investigated (i) whether the prevalence of ticks is higher (or not) in zoo enclosures compared to in the areas surrounding the enclosures and to control areas with similar altitude, vegetation, humidity and temperature, and (ii) whether the presence of *Chlamydia*-related bacteria in ticks may vary according to the environment in which the ticks are collected, i.e., whether they are more common in some enclosures, close to specific animals, and whether they are more likely to be documented in zoo enclosures as compared to areas surrounding the enclosures and/or control neighboring areas. 

## 2. Materials and Methods

### 2.1. Tick Sampling

Ticks were collected from July to September 2023 by flagging low vegetation in six areas throughout the Canton Vaud, Switzerland [38,39] (Figure 1, Table 1): two zoological gardens (Garenne Zoo (GZ) and Servion Zoo (SZ)), two areas contiguous to the zoological gardens, separated by a minimum of three meters from the Zoo fences (Garenne contiguous area (GCA) and Servion contiguous area (SCA)), and two areas sharing similar altitude, vegetations, temperature and hygrometric characteristics compared with zoological gardens, but two to five kilometers away (Garenne control area (GC) and Servion control area (SC)). 

We decided to divide GZ and SZ into three subsections: areas inside the Zoo enclosures (hereafter named “Enclosure”), areas directly outside of enclosures, next to the fence of the enclosure and up to a distance of three meters from fences (named “Surrounding”), and areas within the zoo perimeter but outside the enclosures and outside the surrounding areas (called “Outside”).

No preliminary study was conducted to identify and precisely delimit the contiguous zoo areas and control areas. The zoological gardens were both approximately the same size and were situated at the same altitude, with similar hygrometric characteristics. SZ presented animals that were not present are GZ (Appendix A). In each zoological garden, a maximum flagging time of 200 +/− 10 min was determined prior to the study. For GCA, as well as for GC, SCA, and SC, a maximum flagging time of 260 min was determined. Tick flagging in GCA, GC, SCA and SC was stopped earlier than 260 min, i.e., when about 50 to 60 ticks were collected. For the zoos, the aim was to flag at least once in each enclosure and surrounding area where it was possible to go. Regarding contiguous and control areas, the aim was to investigate as many areas as possible in which ticks could be found. Some areas were flagged several times over the course of the study. 

Sampling of ticks was organized into two-minute sessions of flagging, followed by the examination of the flag. In total, we conducted 122 flagging sessions ranging from two to ten minutes, collecting 212 *Ixodes ricinus* ticks in 1084 min (Appendix A). 

The same white flag of 1 m^2^ was used throughout the whole study (Appendix A). Ticks were identified at the species level based on macroscopic appearance and local epidemiology [9]. We collected larvae, but we did not use them for calculating *p* values, given the possibility of confounding the results [29,40]. They are nevertheless represented in Figure 2, Figure 3, Figure 4 and Figure 5, and reported in the tables (in brackets). 

Ticks were individually transferred into tubes of 2 mL and conserved at −80 °C for about 2 months. Weather was documented using the Swiss weather mobile application and each region was flagged during both good weather and cloudy periods (Appendix A) (MeteoSuisse, available online: https://www.meteosuisse.admin.ch/#tab=forecast-map, accessed on 20 July 2023, 25 July 2023, 26 July 2023, 27 July 2023, 28 July 2023, 3 August 2023, 4 August 2023, 8 August 2023, 9 August 2023, 10 August 2023, 11 August 2023, 15 August 2023, 8 September 2023, 9 September 2023, 11 September 2023 and 13 September 2023). 

### 2.2. Ticks Lysis, DNA Extraction and Pan-Chlamydiae TaqMan Quantitative PCR (pC-qPCR)

Every tick was tested in duplicate. To increase reproducibility and to reduce DNA extraction contamination, the nucleic acid extraction was performed using the automated MagNA Pure 96 instrument (Roche, Penzberg, Germany). Briefly, after ~2 months at −80 °C, ticks were washed with ethanol 70% in order to avoid microbial contamination and transferred to MN bead type E tubes. Ticks were homogenized with Precellys (Bertin technologies, Montigny le Bretonneux, France), diluted with 540 µL ATL buffer and 30 µL of proteinase K, and incubated overnight at 56 °C. Supernatant was then eluted using a 96-well MagNA Pure processing cartridge. A DNA extraction negative control was added to each MagNA Pure extraction run. No inhibition control was used, since we did not observe inhibition when starting from individual ticks. 

After DNA extraction, ticks were screened for the presence of *Chlamydia*-related bacteria using an in-house pan-*Chlamydiae* quantitative PCR (pC-qPCR). The assay amplifies a fragment of the 16S rRNA gene, as described previously [41]. For each PCR run, we used as positive controls, and as a reference for bacterial quantification, 5 microliters of a serial ten-fold dilution of plasmids exhibiting the target DNA sequence of the PCR. Serial dilution ranged from 100,000 to 1 copy/microliter. In addition, for each PCR run, we used a no-template (i.e., no genomic DNA and no plasmid) negative control. 

A tick was considered positive for *Chlamydiaceae* if at least one duplicate had a pC-qPCR ct value ≤ 35. Every sample with a pC-qPCR ct value ≤ 35 was submitted for sequencing (41). Up to two sequences of the 16S rRNA encoding gene were obtained per tick. The obtained short sequences of the 16S rRNA encoding gene were assigned taxonomically on the basis of the genetic distances from the closest sequences available in the NCBI database, i.e., as determined by a basic local alignment search tool (BLAST) [42], using the NCBI database. Practically, we considered that the bacteria present in a sample were part of a given family if the sequence similarity was ≥90% [43]. When the two sequences yielded conflicting results, three additional sequencings were performed (Appendix A).

### 2.3. Geopositioning and Statistical Analysis

After each flagging session (2–10 min), we obtained geographical coordinates using the My GPS Coordinates application on iPhone 7, version 15.3.1. EarthPro, version 7.3 was used to draw Figure 1, Figure 2, Figure 3, Figure 4 and Figure 5. For each session of sampling, ranging from 2 min to 10 min, we calculated the number of ticks obtained per minute of flagging (t/min) (Appendix A). The frequency of ticks detected per minute of flagging was compared using the Wilcoxon rank sum test (Mann–Whitney U test), and the Kruskal–Wallis test. To increase the statistical power, and given the similarities regarding geographical and hygrometric characteristics of both zoos, we then grouped the corresponding areas of La Garenne and Servion together.

For categorical variables, proportions were analyzed using the χ^2^ test, or the two-sided exact Fisher test when conditions for χ^2^ were not met. Statistical analysis was performed using STATA version 17.0.

## 3. Results

First, the mean ticks/min during flagging were not similar between enclosures, the immediate surroundings of enclosures (hereafter called “surroundings”), and outside areas (outside of the enclosures and the immediate surrounding areas, but still inside the zoo), contiguous control areas, and distant control areas (*p* value = 0.0001). Second, the mean ticks/min value was significantly lower in zoological garden enclosures as compared to outsides and surroundings (*p* value = 0.0294), as well as compared to outsides, surroundings and contiguous areas taken together (*p* value = 0.0002), and also as compared to outsides, surroundings, contiguous areas and control areas, all considered together (*p* value < 0.00001). 

The ticks in zoos seem rather to be present in the area surrounding the enclosure, i.e., in the “no man’s land” area where grass is neither cut, nor fouled by zoo visitors, zoo employees, or captive animals, and in areas with grass or bushes in the zoological gardens rather than inside the enclosures, where the vegetation is much limited, in both ruminant and carnivore enclosures (Appendix A).

When comparing GZ to SZ, we observed that the mean ticks/min in SZ and SCA, taken together, was significantly higher than the mean ticks/min of GZ and GCA, taken together (*p* value = 0.0146). However, there was no significant difference in the mean ticks/min for all Servion regions (SZ, SCA, SC) compared to all la Garenne regions (GZ, GCA, GC), because GC had a higher mean ticks/min than SC (*p* value = 0.1083).

The majority of the larvae were collected in SZ (68.2%, 15/22), including 14 during a single flagging session of 2 min (Appendix A).

Among all 190 ticks, 11.58% were positive for *Chlamydiae* (22/190). There was no significant difference in the prevalence of *Chlamydiae* in ticks from zoos (30%, 3/10) versus in ticks from contiguous areas (13.16%, 10/76) (*p* = 0.172, Table 2). There was also no significant difference in infection of ticks from zoos (30%, 3/10) versus ticks from control areas (8.65%, 9/104) (*p* value = 0.0705, Table 2). Comparing the positivity of ticks from zoological gardens with that of ticks from contiguous and control areas, taken together, the difference is not significant (*p* value = 0.0946). 

However, interestingly, the positivity of ticks from outside regions is significantly higher than the positivity of ticks from contiguous and control areas, taken together (*p* value = 0.0126). 

SZ, SCA, and SC had more *Chlamydiae*-positive ticks than GZ, GCA, and GC, but the results were not significant (Table 2).

Interestingly, adult ticks were more likely to be infected by *Chlamydiae* than nymphs (*p* value < 0.00001, Table 3) and larvae (*p* value = 0.005, Table 3).

No new family-level lineage was detected. Among 22 ticks, 72.7% were infected with *Parachlamydiaceae* (16/22), 9.1% with *Rhabdochlamydiceae* (2/22), and 4.5% with *Simkaniaceae* (1/22) (Figure 6). For 13.6% of the positive ticks (3/22), it was not possible to obtain a sequence of sufficient quality to identify the bacteria at the family level (Appendix A); however, these three sequences corresponded to a bacteria belonging to the *Chlamydiae* phylum. Notably, among 16 samples that were positive for *Parachlamydiaceae*, we obtained a single sequence for nine samples and two sequences for the remaining seven samples. Among the samples with two sequences available, two samples had one sequence corresponding to a member of the *Parachlamydiaceae* family and one sequence or which sequencing failed (not identified or not determined, ND), three samples exhibited two different *Parachlamydiaceae* sequences, and one sample had two congruent *Parachlamydiaceae* sequences. Interestingly, one sample was positive for both *Parachlamydiaceae* and *Simkaniaceae*. After additional analysis, we classified this sample as being positive for *Parachlamydiaceae* (see Discussion). Five different sequences of the *Parachlamydiaceae* family were represented. The two ticks infected with *Rhabdochlamydiaceae* each had two sequences available, and these four had >90% similarity with the same 16S rRNA sequence of the family-level lineage of the *Rhabdochlamydiaceae*. The only tick infected with *Simkaniaceae* had two sequences, both exhibiting >90% similarity with the same 16S rRNA sequence of the family-level lineage of the *Simkaniaceae* (Appendix A).

*Parachlamydiaceae* were detected in all six of the different areas (GZ, GCA, GC, SZ, SCA, SC). We detected *Rhabdochlamydiaceae* in SCA and SC. The only tick positive for *Simkaniaceae* was in SZ, Outside (Figure 2, Figure 3, Figure 4 and Figure 5).

*Rhabdochlamydiaceae* presented significantly higher mean numbers of copies/μL than *Parachlamydiaceae* (*p* value = 0.0131) (Appendix A). Such very high titers of DNA for *Rhabdochlamydiaceae* have already been observed in previous studies [23], enabling the direct genome sequencing of the *Rhabdochlamydia helveticae* bacteria [44].

## 4. Discussion

Because zoological gardens would constitute a favorable environment for the development of ticks [24,25,26,27,28], we wondered whether ticks were more often present in zoological gardens than in contiguous areas and/or in distant control areas. Ticks were collected directly from animals in previous studies [27,37,45,46]. The present work is the first, to the best of our knowledge, to collect ticks in zoos and in control areas by flagging, and also the first study to identify *Chlamydia*-related bacteria in ticks collected from zoos. 

Many factors could explain the low prevalence of ticks inside the zoos. First, the enclosures are cleaned regularly, the grass outside the enclosures is regularly cut for the comfort of the visitors, and the grass is also cut inside the enclosures (except when ruminants are present, but then the grass is still rather low due to their intense feeding activity on the grassland; see Appendix A for representative pictures of the enclosures and control areas). Second, the animals are sometimes treated with acaricides [47]. Third, caretakers may treat some animals with antibiotics such as doxycyclin, which might have a detrimental impact on tick populations, if it kills bacterial symbionts beneficial for tick reproduction such as *Wolbachia*. Fourth, enclosures generally consist of pure grassland areas, without the bushes that are suitable habitats for *I. ricinus* [2,3] (Appendix A). Fifth, there are likely fewer free-living rodents in either of the Swiss zoos than in the similar contiguous and control areas outside the zoos, since the carnivores captive in the zoo have a detrimental impact on these free-living rodents, and since the caregivers actively limit the presence of rodents using mouse traps and other mitigation measures. Finally, we focused on questing ticks only, and we did not look at the ticks present on the animals.

Despite being rather unexpected, our results are supported by the study conducted in the Zoological Park Foundation of São Paulo, where only 15 ticks were collected on captive animals, whereas 508 ticks were collected on free roaming animals present in this park [37].

The flagging approach exhibits some limitations, since it tends to select for questing ticks such as *I. ricinus*, which are typically present on herbs and grass, while counterselecting for more aggressive ticks such as *Amblyomma*, which sense the CO_2_ exhaled by mammals and have typical attack strategies [48]. Notably, caregivers were told to provide us any ticks recovered during the study period, but we did not receive any using this complementary approach, either (i) due to poor compliance of caregivers with the study protocol or (ii) due to the excellent health and hygiene of the captive animals. 

However, the present work might better reflect the true prevalence of ticks in zoos than most previous studies, where ticks were collected directly from animals, excluding sampling from potentially dangerous animals. 

The difference in tick frequency between enclosures (inside cages) and the other areas of the zoological garden highlights the possible sampling bias that may occur when studying such a fragmented environment (see below for a discussion regarding positivity of ticks in zoological gardens). It also underlines the importance of precisely recording the sites of sampling. 

We did not expect a difference in tick prevalence between Servion and La Garenne. We hypothesize that differences in humidity, rather than vegetations, temperature or altitude, could explain this difference. Indeed, the Servion zoological garden is centered around a large pond that provides humidity all year long to this zoo (Figure 4). The role of humidity and distance from a water point has already been shown to be very important in predicting the presence/absence of *Ixodes ricinus* ticks [5]. 

The positivity of ticks corresponds to previous results from Switzerland, where the prevalence ranged from 0.89% to a maximum of 28.4% [9,23]. This is much higher than the prevalence in Swiss ticks of most tick-borne agents, such as *Anaplasma*, *Babesia*, *Rickettsia*, *Francisella* and TBEV, the prevalence of all of which ranges from 0 to 3% [38,49,50,51], and this provides additional hints for the role of ticks as reservoirs of *Chlamydia*-related bacteria [9].

The prevalence of *Chlamydiae* was significantly higher in adults, and none of the larvae were positive for *Chlamydiae*. This makes sense, assuming the absence of transovarial transmission of these bacteria in *Ixodes ricinus* ticks [9]. This might indicate that nymphs and adult ticks become progressively infected when feeding on mammals, birds and/or reptiles.

Conducting the analysis from specific areas inside the zoo might explain the discrepancy with studies in which the same prevalence of tick-borne pathogens was observed inside and outside zoological gardens [29].

Unexpectedly, *Chlamydiae*-infected ticks were only found in the zoo areas outside of the enclosures, where the restaurants, toilets and children’s playgrounds are located. This may reflect an association of *Chlamydia*-related bacteria with certain human-activity-related products such as food, man-made water systems and garbage, which are typically associated with cockroaches and free-living amoebae. Cockroaches are known to be a reservoir for *Rhabdochlamydiaceae* [52], whereas *Parachlamydiaceae* are largely associated with amoebae [53,54].

Our results confirm that *Parachlamydiaceae* and *Rhabdochlamydiaceae* are the most represented families in Swiss ticks [9,23]. This was expected, since *Rhabdochlamydiaceae* has already been found previously in captive animals [34], and has often been documented in ticks [9,23]. Interestingly, the prevalence of *Rhabdochlamydiaceae* was lower compared to that reported in previous studies performed in Switzerland using a very similar methodology [9,23]. The diversity of *Chlamydia*-related bacteria observed was also lower than in previous studies, possibly due to the “cleaner” environment and usage of acaricides and antibiotics to treat some of the captive animals. Previous studies analyzing *Chlamydia*-related bacteria among Swiss ticks have always detected some unclassified members of the *Chlamydiales* order [9,23]. Here, we failed to discover new family-level lineages, likely because of the limited number of ticks investigated (only 212, as compared to 56,000 in the work by *Pilloux* et al. [9]).

No *Chlamydiaceae* were found, which is similar to what was observed in a survey performed on animals from a zoo in Japan [55]. 

Interestingly, some ticks presented *Chlamydiae* from different strains assigned to the *Parachlamydiaceae* family (Appendix A). This could either (i) suggest that some ticks can be infected with different species of the *Parachlamydiaceae* family or (ii) correspond to a limitation of the BLAST analysis performed on the minimally discriminative 16S rRNA encoding genes sequences [17].

Zoological gardens might represent a favorable environment for ticks to acquire *Chlamydiae* bacteria. However, the limited number of ticks found in both zoological gardens prevents us from drawing firm conclusions regarding *Chlamydiae* prevalence and diversity, even if the prevalence of infected ticks seems to be higher, especially in the area distant from the enclosures (outside), rather than inside the cages. 

We are confident that our results are robust. First, the total absence of members of the *Chlamydiaceae* family speaks against any cross contamination of samples during processing, since in our medical university hospital laboratory, *C. trachomatis* and other *Chlamydiaceae* are the lineages most frequently amplified from human samples. Second, all negative no-template controls were negative and all negative DNA extraction controls were negative, whereas the positive controls were systematically positive when at least 10 copies of target DNA/microliter were used. Moreover, we developed this PC-qPCR in Lausanne, and we used it regularly for a wide diversity of projects with no evidence of amplicon contamination. Rather, we estimated from past projects that a PCR contamination in our TaqMan assay would occur in fewer than 1 per 10,000 PCRs, if any [56,57,58]. Thus, we amplified no DNA from *Coxiella* when performing more than 2000 PCRs in blood donors and more than 8000 PCRs on tick pools [56], despite the TaqMan Coxiella PCR being highly sensitive, with an analytical sensitivity of 10 copies/microliter, and amplifying >80% of the positive human heart valve samples studied blindly [57]. This high specificity is likely due to the fact that with TaqMan real-time PCRs, we do not open the tube to detect amplicons, and due to the fact that these PCRs—even when part of master’s or PhD thesis projects—are systematically performed by trained laboratory technicians, who have been trained to follow the quality assurance protocols implemented in our diagnostic-accredited laboratory. 

Additional analyses were conducted for two samples. Sample number 11.3.3 was slightly positive, with a Ct value of 34,935, whereas in the duplicate, it was negative (Appendix A). This result was compatible with a false positive, and was thus retested three more times, with all subsequent results being negative. Sample number 5.1.1 exhibited discrepant sequencing results (Appendix A). These results could be explained by the co-occurrence in the same tick of two strains belonging to two different families (*Parachlamydiaceae* and *Simkaniaceae*) or sequencing errors. Additional analysis confirmed the presence of the *Parachlamydiaceae* family only. This highlights the importance of conducting analyses in duplicates, while also showing the limited discriminative power of 16S rRNA sequences-based classifications [17].

Unfortunately, we were not always able to conduct all of the flagging sessions for each area, for the following reasons: (i) caregivers sometimes had to put the animals back into their enclosure after only 2 min; (ii) changes in weather or the presence of brambles at the sampling site forced us to shorten the flagging time of some sessions outside the enclosures (Appendix A). This may have introduced bias, overestimating the absence of ticks in some enclosures. However, the frequency of ticks retrieved per minute was considered, and mean flagging time was similar overall in the different areas (Appendix A). A second limitation of our work is that flagging was performed neither during the most active period (Spring) for ticks nor throughout the entire year. Nevertheless, this limitation is partially mitigated by using controls during the same period. 

## 5. Conclusions

In conclusion, ticks can be detected in zoos in Switzerland, mainly outside the enclosures, and they may be infected with *Chlamydia*-related bacteria. We also showed a predominance of *Parachlamydiaceae* in ticks, and to a lesser extent the presence of *Rhabdochlamydiaceaee* and *Simkaniaceae*. Moreover, our data also suggest an impact of humidity on the presence of ticks. Our work highlights the importance of performing sub-analysis in which larvae are taken into account (or not), and the relevance of separating the sampling of ticks in an environment into different areas with different characteristics. Further studies sampling ticks in zoological gardens should concentrate their priority on areas with bushes or tall grass. Considering the potential threat to human and animal health, the impact of zoos on the presence and diversity of *Chlamydiae* among ticks needs to be further investigated. 

## Figures and Tables

**Figure 1 microorganisms-11-02468-f001:**
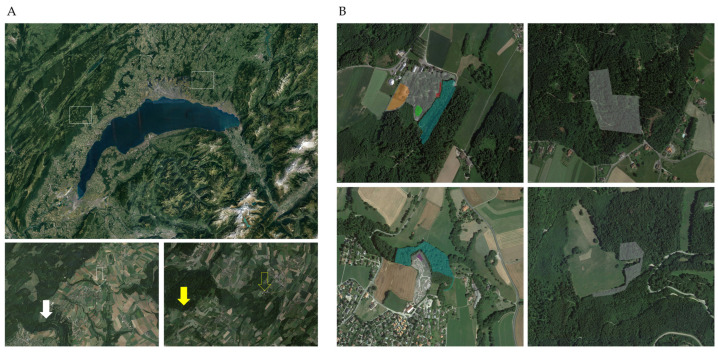
**Organization of the different sampling areas.** (**A**) The upper panel shows an aerial view of the region of Leman lake, Switzerland. The white rectangles indicate the areas corresponding to the Garenne Region (the **left** rectangle) and the Servion Region (the **right** rectangle). In the left lower panel, the full white arrow indicates the Garenne control area (GC), and the empty arrow highlights the Garenne contiguous area (GCA) and Garenne Zoo (GZ). In the right lower panel, the full yellow arrow indicates the Servion control area (SC), and the empty yellow arrow highlights the Servion contiguous area (SCA) and Servion Zoo (SZ). (**B**) Aerial view of the six main sampling areas: the upper left panel corresponds to Servion Zoo (SZ, in white) and its immediate contiguous area (SCA, in green); the lower left panel highlights Garenne Zoo (GZ, in white) and its immediate contiguous area (GCA, in green); the right upper and lower panels show the Servion and Garenne control areas (SC and GC, respectively). For further explanations, see Figure 2, Figure 3, Figure 4 and Figure 5.

**Figure 2 microorganisms-11-02468-f002:**
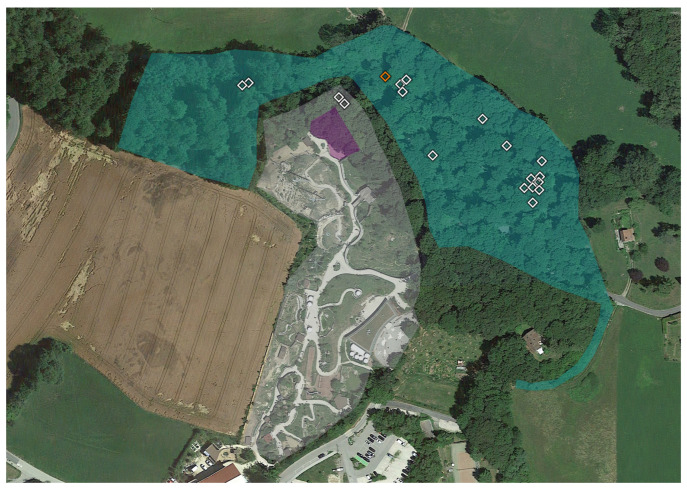
**Aerial view of Garenne Zoo (highlighted in white, GZ) and the contiguous area (in green, GCA).** The wild pig enclosure is highlighted in purple, around which we found two ticks. The shapes indicate the life stage and (for adults) sex of the ticks: diamonds nymphs. Ticks with *Chlamydiae* DNA are highlighted in color. Orange corresponds to *Parachlamydiaceae*.

**Figure 3 microorganisms-11-02468-f003:**
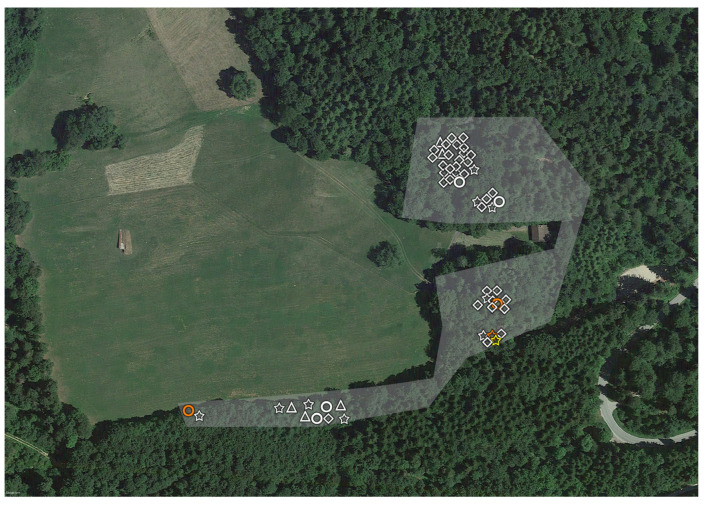
**Aerial view of the Garenne control area (highlighted in white, GC) located at about 2–5 km from Garenne Zoo (GZ).** The shapes indicate the life stage and sex of the ticks: triangles indicate larvae, diamonds nymphs, circles females and stars males. Ticks with *Chlamydiae* DNA are highlighted in color. Orange corresponds to *Parachlamydiaceae*, and yellow to *Chlamydiae* sequences that could not be assigned at the family level (ND, not determined).

**Figure 4 microorganisms-11-02468-f004:**
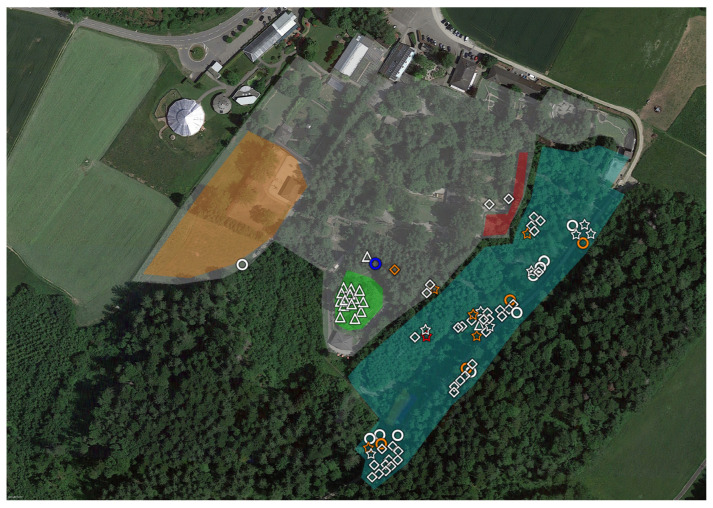
**Aerial view of Servion Zoo (highlighted in white, SZ) and the contiguous area (in green, SCA).** The enclosure of snow leopards is highlighted in green, from which 14 larvae were collected. Red indicates the reindeer enclosure, from around which two ticks were collected. Orange indicates the buffalo enclosure, from around which one tick was collected. The shapes indicate the life stage and sex of the ticks: triangles indicate larvae, diamonds nymphs, circles females and stars males. Ticks with *Chlamydiae* DNA are highlighted in color. Orange corresponds to *Parachlamydiaceae*, red to *Rhabdochlamydiaceae*, and blue to *Simkaniaceae*.

**Figure 5 microorganisms-11-02468-f005:**
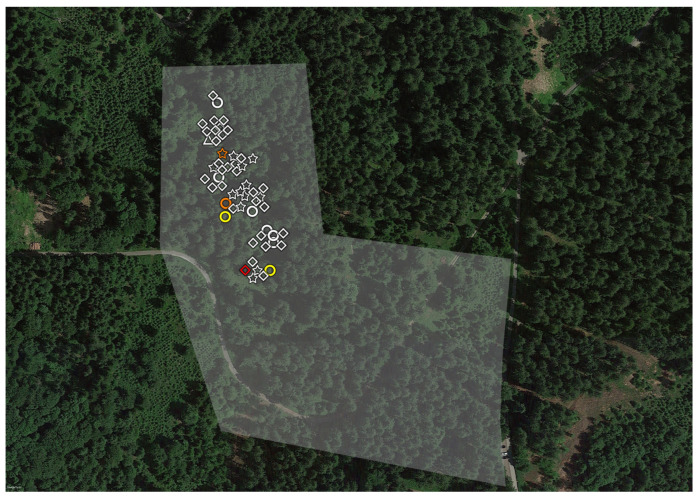
**Aerial view of the Servion control area (highlighted in white, SC) located about 2–5 km from Servion Zoo (SZ).** The shapes indicate the life stage and sex of the ticks: triangles indicate larvae, diamonds nymphs, circles females and stars males. Ticks with *Chlamydiae* DNA are highlighted in color. Orange corresponds to *Parachlamydiaceae*, red to *Rhabdochlamydiaceae*, and yellow to *Chlamydiae* sequences that could not be assigned at the family level (ND, not determined).

**Figure 6 microorganisms-11-02468-f006:**
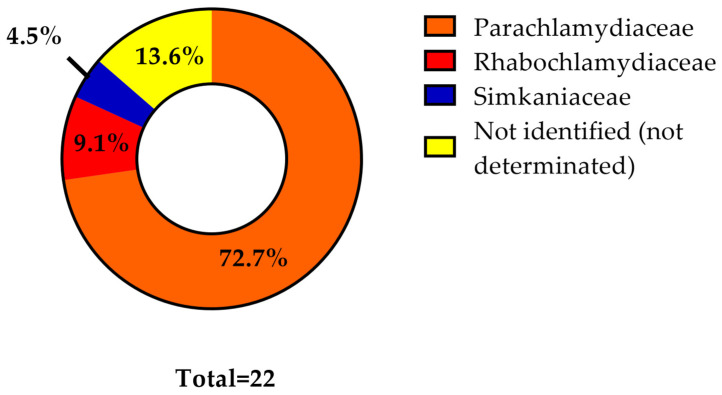
**Distribution of the obtained Chlamydia-related bacteria, according to family-level assignment.** The high prevalence of *Parachlamydiaceae* (72.7%) confirms the results of previous studies, but the low prevalence of *Rhabdochlamydiaceae* (9.1%) was rather unexpected.

**Table 1 microorganisms-11-02468-t001:** **Locations at which the 212 *Ixodes ricinus* ticks collected in this work were documented.** In parentheses, the number is shown while taking into account larvae. Please note that more than half of larvae were collected in SZ.

Place	Ticks Servion	Ticks Garenne	Ticks Garenne and Servion
Zoo	8 (23)	2 (2)	10 (25)
-Enclosure	0 (14)	0 (0)	0 (14)
-Surrounding	3 (3)	2 (2)	5 (5)
-Outside	5 (6)	0 (0)	5 (6)
Contiguous area	59 (60)	17 (17)	76 (77)
Control area	54 (55)	50 (55)	104 (110)

**Table 2 microorganisms-11-02468-t002:** **Prevalence of members of the Chlamydiae phylum among ticks according to the studied area.** Please note that for every location, the number of positive ticks is higher in Servion compared to La Garenne. Larvae are not represented.

Place	Chlamydia Prevalence in Ticks from Servion (%) ^1^	Chlamydia Prevalence in Ticks from Garenne (%) ^1^	Chlamydia Prevalence in Ticks from Garenne and Servion (%) ^1^
Zoo	37.5 (3/8)	0 (0/2)	30 (3/10)
-Enclosure	0 (0/0)	0 (0/0)	0 (0/0)
-Surrounding	0 (0/3)	0 (0/2)	0 (0/5)
-Outside	60 (3/5)	0 (0/0)	6 (3/5)
Contiguous area	15.25 (9/59)	5.88 (1/17)	13.16 (10/76)
Control area	9.26 (5/54)	8 (4/50)	8.65 (9/104)

^1^ Number of ticks positive with the pC-qPCR divided by the total number of ticks tested by pC-qPCR.

**Table 3 microorganisms-11-02468-t003:** Prevalence of *Chlamydia*-related bacteria DNA in 212 *Ixodes ricinus* ticks according to developmental stage and location.

Tick Developmental Stage	Number of Ticks	Enclosure	Surrounding and Outside	Contiguous Area	Control Area	*Chlamydiae* Prevalencein Ticks (%) ^1^
Larvae	22	14	1	1	6	0 (0/22)
Nymphs	122	0	7	50	65	2.46 (3/122)
Female adults	30	0	2	13	15	30 (10/30)
Male adults	38	0	1	13	24	23.68 (9/38)
Total	212	14	11	77	110	10.38 (22/212)

^1^ Number of ticks positive with the pC-qPCR divided by the total number of ticks tested by pC-qPCR.

## Data Availability

All data are presented in the main text and in the Appendix A.

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
