# Peer review of "Ticks and Chlamydia-Related Bacteria in Swiss Zoological Gardens Compared to in Contiguous and Distant Control Areas"

_microorganisms, 2023, doi:10.3390/microorganisms11102468_

Round 1
Reviewer 1 Report
The article is about the prevalence of ticks and Chlamydia-related bacteria in ticks in Swiss zoological gardens.
Introduction
Line 74: Thus, we wondered "we investigated"
Material and methods
Tick sampling
Line 84-95: Please check the formatting of text.
Please add references in tick sampling. Line 84-122: There is no reference.
Tick lysis-----
Please add detail about quality and quantity checks in DNA extraction part.
There is no detail of positive control in qPCR.
Results
Line 161-162: This is the part of discussion.
Line 164-171: It should be included in methodology.
Results should not include methodology and discussion. Please proofread and re-write results.
In methods line 144-145 "positive samples were then sequenced as reported (39). Identification at the family level was made by basic local alignment search tool (BLAST) [40], using the NCBI database". Kindly explain phylogenetic analysis, I could not find phylogenetic tree in results. However, data is present in supplementary table.
Discussion is too long.
Minor editing is required.
Reviewer 2 Report
The authors present an interesting and novel approach towards investigation of distribution of ticks in Zoos and surrounding areas as well as identification of Chlamydiae in the ticks. Two Zoos in Canton Vaud, Switzerland, served as model sites. The results of the study have a potential to be published. However, the manuscript in its present form is not suitable for acceptance and major revision is necessary.
One of the major drawbacks is the language and style. The text is too wordy and repetitive and should be reduced in extent by at least 40%. The text needs thorough and in depth proof reading of the language with the help of a native speaker, preferably expert in the field.
Specific comments
My major concern is the choice of the flagging period. Flagging in the period of the highest seasonal activity of I. ricinus, or during the whole season should have been done. Please explain the rationale for choosing July to September for collecting ticks from vegetation (i.e. obviously the period after the peak seasonal activity) and discuss the limitations of the study caused by this choice.
Discussion is too long and contains repetition of the results. Please make it more concise and comprehensive.
The conclusions of the study are based on two zoos and a limited flagging period thus please be more careful in interpretation and generalization of the results.
Minor comments
Please improve the figures by increasing the size of the labels. Some of them seem to be missing (see more comments in the attached file)
Format of the list of references should strictly follow submission guidelines of the journal.
More specific comments and a few corrections (but by far not all) are included in the attached manuscript file.

The text is too wordy and repetitive and should be reduced in extent by at least 40%. The text needs thorough and in depth proof reading of the language with the help of a native speaker, preferably expert in the field.
Round 2
Reviewer 1 Report
Authors complied with almost all comments.
Kindly proofread manuscript carefully.
Reviewer 2 Report
The manuscript has been improved, but is still not ready for acceptance.
The extent is even larger than that of the original version. This way it is too wordy and repetitive.
Please reduce the text by 1/ avoiding repetitions of the methods in the results and discussion, 2/ deleting the newly added results from the methods (i.e. L. 128-132), 3/ avoiding repetitions of the results in the discussion.
It is not necessary to repeat all the time that the results were obtained in your study, nor what PCR methods were used, etc.
The language was corrected to some extent, but many errors are remaining. See just a few examples, but by far not all (line numbers refer to the clean version):
L- 71 – Sao Paulo, instead of Sao Polo
L. 158-159 correct to ... we did not observe ...
L. 185 correct to „...statistical...“
L. 198 should probably be „called“ instead of „coined“
L. 274 correct to „significant“
L. 374 correct to „the study“
L. 421-423 correct to ...“ they did not conduct any ..... discrepancy...“
L. 429 correct to „cockroaches“
L. 440-441 correct to ... usage of acaricides & antibiotics to treat some …
L. 450 correct to „ family....“
L. 478 ....“ is likely due to „...
L. 486 „ determined“
L. 489 „ discrepant“
L. 497 „in duplicates“
L. 527 „neighboring“
Thus, the text needs another thorough and in depth proof reading of the language.
In figure 1, explanations of some colours are missing. Although they are explained in the next figures, please include at least references to those figures, e.g. „for further explanations see Figures 2 and 4.
The explanation why the authors did not increase the size of the labels in the figures 2-5 is not acceptable. Imagine that the quality of the figures will be lower in the printed paper than in the submitted figures and the readers will not be able to identify the labels at all.
Table 2: please add the numbers of examined ticks and prevalence, e.g. in the form
X% (3/Y), i.e. prevalence in % (number of positive/number of examined ticks)
Conclusions should be more concise.
The language was corrected to some extent, but many errors are remaining. The text needs another thorough and in depth proof reading of the language.
